# Ultrasonic Diagnosis and Management of Posthemorrhagic Ventricular Dilatation in Premature Infants: A Narrative Review

**DOI:** 10.3390/jcm11247468

**Published:** 2022-12-16

**Authors:** Gengying Liu, Chuan Nie

**Affiliations:** 1Neonatology Department, Guangdong Women and Children Hospital, Guangzhou 510010, China; 2Guangdong Neonatal ICU Medical Quality Control Center, Guangzhou 510010, China

**Keywords:** preterm infant, intracerebroventricular hemorrhage, posthemorrhagic ventricular dilatation, ventricular access device, ventriculoperitoneal shunt

## Abstract

The survival rate of preterm infants is increasing as a result of technological advances. The incidence of intraventricular hemorrhages (IVH) in preterm infants ranges from 25% to 30%, of which 30% to 50% are severe IVH (Volpe III-IV, Volpe III is defined as intraventricular bleeding occupying more than 50% of the ventricular width and acute lateral ventricle dilatation, Volpe IV is defined as intraventricular hemorrhage combined with venous infarction) and probably lead to posthemorrhagic ventricular dilatation (PHVD). Severe IVH and subsequent PHVD have become the leading causes of brain injury and neurodevelopmental dysplasia in preterm infants. This review aims to review the literature on the diagnosis and therapeutic strategies for PHVD and provide some recommendations for management to improve the neurological outcomes.

## 1. Introduction

Due to the regional vulnerability of the subependymal germinal matrix and abrupt fluctuations in cerebral blood flow, preterm infants, particularly very premature infants, are prone to developing an intraventricular hemorrhage (IVH). Bleeding usually occurs in the germinal matrix near the ventricular wall and is rich in an immature capillary network. An IVH is also called a germinal matrix hemorrhage-intraventricular hemorrhage (GMH-IVH). Blood cells from an IVH may block Montessori’s foramen, resulting in obstructive hydrocephalus. Moreover, secondary inflammation caused by a hematoma and scarring of the arachnoid villi would bring a cerebrospinal fluid resorption disorder, which would lead to communicating hydrocephalus and eventually to posthemorrhagic ventricular dilatation (PHVD). Studies have shown that between 30% and 50% of severe IVH (Volpe III–IV, Volpe III is defined as the intraventricular bleeding occupying more than 50% of the ventricular width and there being an acute dilatation of the lateral ventricles, Volpe IV is defined as an intraventricular hemorrhage combined with venous infarction [1,2]) cases would finally entail PHVD, which has become one of the main reasons of neurodevelopmental dysplasia and increased mortality in preterm infants [3,4,5,6,7,8]. Yet, there is no consensus on treatment protocols for PHVD, and diagnostic approaches for PHVD appear to differ from one neonatal center to another. It was usual practice to use cranial ultrasound (cUS) to measure the size of the ventricles to diagnose PHVD, but there is no agreement on which ventricular characteristics should be increased and to what degree [9]. Some facilities advocated early intervention, whereas others started intervention when symptoms of increased intracranial hypertension emerged [10]. This narrative review aims to explore the brief literature on the ultrasound diagnosis and therapeutic strategies of PHVD to provide some recommendations for management.

## 2. Ultrasound Diagnosis of PHVD

PHVD usually occurs on the seventh–fourteenth days after an IVH. cUS is a good measurement for diagnosing and following the progression of PHVD according to these three parameters [11,12]: (1) The most important and common index is the ventricular index (VI), also known as the Levene index, that is the distance between the cerebral falx and the lateral wall of the anterior horn in the horizontal coronal view of the interocular foramen (Monro foramen) (Figure 1). (2) Anterior horn width (AHW), the diagonal width of the anterior horn measured at the widest part of the horizontal coronal plane of the Monro foramen (Figure 1). (3) Thalamo-occipital distance (TOD), the distance between the outermost end of the thalamus at the junction of the thalamus and the choroid plexus, and the outermost end of the occipital angle (posterior angle) in the parasagittal plane (Figure 2). The criteria for progressive PHVD [13] are: (1) VI > 97th centile + 4 mm of infants with same postmenstrual age the 97th and 97th + 4 mm lines of VI at different gestational ages, as shown in Figure 3, or (2) AHW > 4 mm (>97th centile +1 mm) and TOD > 26 mm (97th centile +1 mm).

Currently, there is no consensus on the best time for the intervention of PHVD [14]. Levene et al. [9] reported that intervention should be performed when VI exceeds the line of 97th centile + 4 mm of infants with the same postmenstrual age (Figure 3 show the 97th and 97th + 4 mm lines of VI at different gestational ages). Experts from Brigham and the Women’s Hospital of Harvard University and other hospitals developed an expert consensus in 2020, dividing infants with PHVD into low-, medium-, and high-risk groups [15]. (1) Low risk: VI < 97th centile, AHW ≤ 6 mm, and no head circumference increase of ≥ 2 cm per week, no cranial suture separation, and anterior fontanel swelling. (2) Medium risk: VI > 97th centile, AHW > 6 mm, or TOD > 25 mm, and no head circumference increase of ≥2 cm per week, also no cranial suture separation and anterior fontanel swelling. Lumbar punctures may be performed two to three times, and monitoring with continuous cranial ultrasound two to three times weekly should take place until there is no evidence of dilatation progression. After two weeks, ultrasound monitoring may be performed once every one–two weeks until 34 weeks postmenstrual age. If dilatation is progressive, neurosurgical intervention should be considered. (3) High risk: VI > 97th centile + 4 mm, AHW > 10 mm, or TOD > 25 mm, or any of the above three symptoms appear, the intervention of lumbar puncture two to two times or neurosurgical intervention should be performed. It has been reported that if the intervention was performed after clinical symptoms had already manifested, up to 92% of patients would eventually require a ventriculoperitoneal shunt (VPS) [5]. However, because of the compliant skulls, large extracerebral spaces, and high water contents of the white matters in preterm infants, it can be challenging to identify the above three symptoms, especially in the early stage. Although measuring the circumference of the head is a simple and inexpensive procedure that should be performed everywhere, some neuroscientists have proposed that there is no correlation between the rapid increase in head circumference and the progression of ventricle dilation in premature infants following intraventricular hemorrhage [16,17].

## 3. Treatment of PHVD

### 3.1. Continuous Lumbar Puncture

A continuous lumbar puncture is the most straightforward treatment for PHVD. It can clear bloody cerebrospinal fluid, reduce cerebrospinal fluid’s proteins and inflammatory factors, and relieve intracranial hypertension. It may be effective for infants with minimal bleeding and mild dilatation. De Vries et al. [6] conducted an ELVIS study that included 126 preterm infants with PHVD whose gestational ages were ≤34 weeks. In this study, the candidates were divided into a low-threshold group (VI > 97th centile, AHW > 6 mm or TOD > 25 mm) and a high-threshold group (VI > 97th centile +4 mm, AHW > 10 mm), and both groups received continuous lumbar punctures daily. As a result, the VPS rate of the low-threshold group was 19% (12/64), and of the high-threshold group was 23% (14/62). There were no statistical differences in the final VPS rates or mortalities between these two groups. This suggests that the therapeutic effect of lumbar punctures at a high-threshold VI > 97th + 4 mm is almost equivalent to that at a low-threshold VI > 97th, but using a high point for lumbar puncture timing can prevent some infants with an unplanned stop-of-progression from receiving an invasive puncture. De Vries et al. [6] reported that dilatation stopped progressing in 25% of cases after two to three lumbar punctures, indicating that the effective rate of lumbar punctures is 25%. However, it has also been reported that continuous lumbar punctures can only temporarily relieves the symptoms of intracranial hypertension and cannot slow the progression of PHVD. There is no evidence that repeated lumbar punctures produce any benefit over conservative management in neonates in terms of reduction of disability, death, or need for the placement of a permanent shunt [18].

### 3.2. Drainage Intervention Fibrinolytic Therapy (DRIFT)

Drainage intervention fibrinolytic therapy (DRIFT) involves inserting a catheter into the ventricle and injecting recombinant tissue plasminogen activator (RT-PA), after which artificial cerebrospinal fluid is used, repeatedly flushing the ventricle until the drainage-recovered fluid is not colored [19]. The purpose of DRIFT is to remove harmful substances such as pro-inflammatory cytokines, free iron, and blood components from the ventricle and reduce secondary damage to brain cells. Whitelaw et al. [13] randomly divided 70 PHVD preterm infants born at gestational ages of 24 to 34 weeks into two groups: one group was treated with DRIFT and the other with standard treatments (lumbar punctures or treatment using a ventricular access device), finding out that there was no statistically significant difference in the VPS rates and mortalities between these two groups. However, compared with the standard treatment group, the rates of severe disability and mortality in the DRIFT group were significantly reduced after two years of follow-up [20]. After ten years of follow-up, the cognitive quotient (CQ) of the DRIFT group was 69.3, which was significantly higher than that of the standard-treatment group (53.7) [21]. This result indicated that DRIFT positively affects long-term cognition, but does not improve motor function. The authors speculated that DRIFT could reduce secondary neurotoxicity and damage to the cerebral cortex but could not promote the regeneration of key motor bundles after severe cerebral hemorrhagic infarction. Park et al. [22] also reported that intraventricular drainage combined with urokinase injections every 3 to 6 h could reduce the VPS rate and improve the neurological prognosis. However, it has been verified that DRIFT can improve the long-term cognitive ability of preterm infants with PHVD, which is beneficial for long-term neurological outcomes, but also can bring infection, secondary bleeding, and rapid fluctuations of intracranial pressure. Whitelaw’s above study also found that 35% of infants treated with DRIFT developed a recurrent IVH, which was higher than the 8% in infants managed with standard treatment.

### 3.3. Ventricular Access Device (VAD)

When continuous cranial ultrasound monitoring indicates progressive dilation or the ventricle dilation does not stop after two to three lumbar punctures, a VAD is often temporarily placed to drain cerebrospinal fluid [23]. The most common VAD device is the Ommaya reservoir [24]. The Ommaya reservoir is a VAD used for repetitive access to the intrathecal space. It consists of an indwelling ventricular catheter with a collapsible silicone reservoir port. The distal end of the catheter is surgically placed into the ipsilateral anterior horn, with the proximal end connected to the reservoir [25]. Ommaya reservoir insertion can not only reduce the frequency of lumbar punctures, but it can also repeatedly extract cerebrospinal fluid based on intracranial pressure. When there are manifestations of intracranial hypertension such as a bulging fontanel, head circumference growth > 2 cm/week, skull suture separation, feeding difficulty, and apnea, a VAD is the first choice, and the drainage volume can be increased to 15 mL/kg/day [15]. It is noteworthy that a rapid decrease of VI should be avoided to avoid secondary bleeding caused by violent intracranial hemodynamic changes. Peretta et al. [26] investigated 17 preterm infants with PHVD after implanting an Ommaya reservoir and reported that it could reduce VPS’s dependence. Lin et al. [27] reported that 3 to 5 weeks after implanting an Omamaya reservoir in 15 infants, the levels of protein, glucose, and red blood cells in their cerebrospinal fluids returned to normal. After a follow-up of 18 to 36 months, one infant required a VPS; one had died, two developed spastic paralyzes of both lower limbs, and another eleven did not have any complications (73%). However, some studies have reported the opposite results. Richard et al. [28] studied 64 infants with PHVD treated with an Ommaya reservoir, and after six months to four years of follow-up, the final VPS rate was 69%. The incidence of severe sequelae was 39%, suggesting that the Ommaya reservoir does not provide good outcomes with respect to mortality, VPS rate, and neurological function.

### 3.4. Ventriculosubgaleal (VSG)

The subgaleal space is the fibroareolar layer of the scalp between the galea aponeurotica and the periosteum of the cranial bones. Due to its elastic and absorptive capabilities, it can be used as a shunt to drain excess cerebrospinal fluid from the ventricles. A VSG consists of a shunt tube with one end in the lateral ventricles and another inserted into the subgaleal space [29]. The placement of the drainage tube is simple, and could even be completed in the neonatal intensive care unit (NICU). Sil et al. [30] found that a VSG could delay or even avoid the placement of a VPS, according to a retrospective study in 2020, effectively reducing the dependence on a permanent shunt. A multi-center study conducted in the United States showed that 31 of 36 (86%) preterm infants treated with a VSG required a VPS, while 61 of 88 (69%) treated with a VAD needed a VPS [31]. The difference was statistically significant, indicating that a VAD’s effect is better than that of a VSG. Notably, there was no significant difference in the incidence of infection between these two groups. However, another study indicated that infection and shunt blockage were the most common complications, and *Staphylococcus aureus* and *Staphylococcus epidermidis* were the most common pathogens [29]. Both VAD and VSG are temporary drainage measures for PHVD. Compared with VAD, VSG reduces the daily aspiration of cerebrospinal fluid [14], and cerebrospinal fluid is resorbed through the space under the galea aponeurotica. No additional puncture or aspiration are required, and the time the device can be used is prolonged. The longest time a VSG was used is reported to be 2.5 years [32]. Fountain et al. [33] systematically reviewed the literature on the outcomes of a VAD and VSG and found no statistical differences in infection rates, catheter blockage rates, VPS rates, or mortalities.

In addition to VAD and VSG as temporary interventions, endoscopic third ventriculostomy (ETV) combined with choroids plexus cauterization [34,35], the intraventricular or venous infusion of bone marrow mesenchymal cells [36,37,38], and the iron chelator deferoxamine have been shown to prevent the progression of PHVD [39]; all of these methods require further study before being widely used clinically, however.

### 3.5. Ventriculoperitoneal Shunt (VPS)

A VPS device comprises a ventricular catheter connected to a valve and a distal catheter in the peritoneal cavity. It is a cerebral shunt that can transfer excess cerebrospinal fluid from the lateral ventricles into the peritoneum when the normal outflow is blocked or there is a decrease in fluid absorption [40]. A VPS is usually the first choice for the treatment of PHVD and hydrocephalus in adults [41]. However, preterm infants, especially extremely low birth weight infants, are prone to infections due to poor immune function. Furthermore, it is common for mechanical obstructions caused by the inadequate peritoneal development and the insufficient absorption capacity of preterm infants. The protein in cerebrospinal fluid is high in the acute stage [42,43]. In addition, evidence suggests that roughly one-third of newborns experienced spontaneous resolution of ventricular dilatation [44]. For these reasons, it is recommended to postpone VPS in preterm infants [45]. VPS is considered a radical treatment for PHVD. A VPS transfers the excess cerebrospinal fluid from the ventricle to the peritoneal cavity, where it is absorbed. However, there are strict indications for the placement of a VPS. (1) After a lumbar puncture, the implantation of a VAD, or the drainage of cerebrospinal fluid by a VSG, the ventricle still shows a progressive dilation, and after four weeks the continuous drainage of cerebrospinal fluid is required to maintain VI < 97th centile + 4 mm. (2) Weight > 2 kg. (3) Cerebrospinal fluid protein < 1.5 g/L. (4) Red blood cell count of cerebrospinal fluid < 100/mm^3^ [46]. Although VPS is an effective treatment, it also increases the risk of infection in preterm infants [47]. A multi-center study performed in the United States and Canada indicated that the infection rates of a VPS placement during the first year ranged from 8% to 10% [48]. Infection and device malfunction are the most critical complications [49,50]. The symptoms of infection included abdominal pain, positive peritoneal irritation sign, and fever. If those who carry the shunt device develop a persistent fever, they are most likely infected. Additionally, using antibiotics alone usually has no impact, necessitating the removal of the implanted shunt device.

## 4. Conclusions

Based on the therapeutic experience from many clinical institutions, preterm infants, particularly very premature infants with gestational ages < 32 weeks, with severe IVH, should be assessed with cUS twice a week until PHVD stops aggravating. An intervention should be performed when VI > 97th centile + 4 mm, AHW > 10 mm, or TOD > 25 mm. When cerebrospinal fluid drainage is required for more than four weeks, and dilation is still in the aggravated stage, a VPS should be placed if the body weight of the infant is >2 kg, the protein of cerebrospinal fluid is <1.5 g/L, and the red blood cell count of cerebrospinal fluid is <100/mm^3^. Based on the reported therapy experiences, we summarized a flow chart (Figure 4) for the management of PHVD to serve as a reliable benchmark. There are significant limitations, such as the fact that almost all research described only evaluated neurological outcomes in terms of basic daily functioning abilities. More additional studies with long-term follow-ups may be required to analyze cognitive processes. Multi-center, large-scale, randomized studies are desired for the improved management of PHVD, and this will be our next effort to find the best treatment regimens to improve the neurological outcomes.

## Figures and Tables

**Figure 1 jcm-11-07468-f001:**
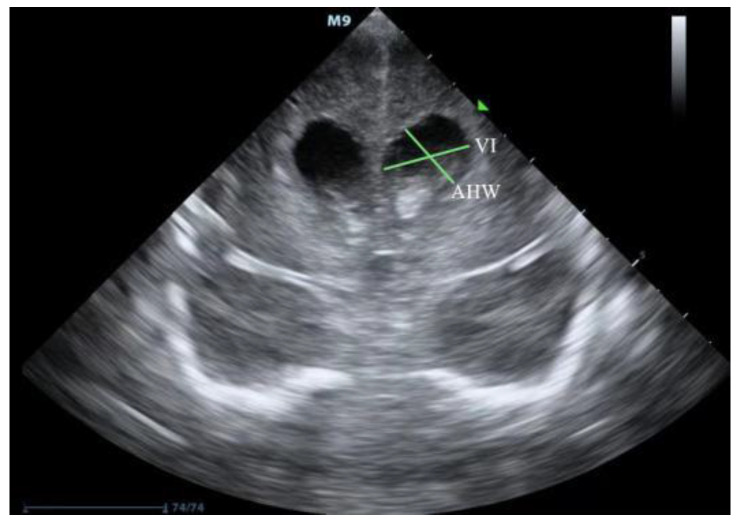
Illustration of measurement of VI and AHW (source: from a very premature infant with gestational age of 30^+3^ weeks and diagnosed with IVH and PHVD in the department of neonatology, Guangdong Women and Children Hospital. IVH: intraventricular hemorrhage, PHVD: posthemorrhagic ventricular dilatation.VI: ventricular index, AHW: anterior horn width).

**Figure 2 jcm-11-07468-f002:**
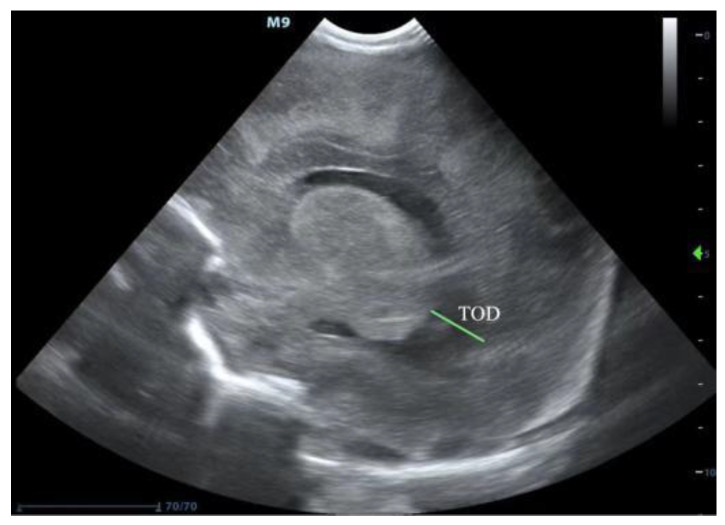
Illustration of TOD measurement (source: from a very premature infant with gestational age of 30^+3^ weeks and diagnosed with IVH and PHVD in the department of neonatology, Guangdong Women and Children Hospital. IVH: intraventricular hemorrhage, PHVD: posthemorrhagic ventricular dilatation, TOD: thalamo-occipital distance).

**Figure 3 jcm-11-07468-f003:**
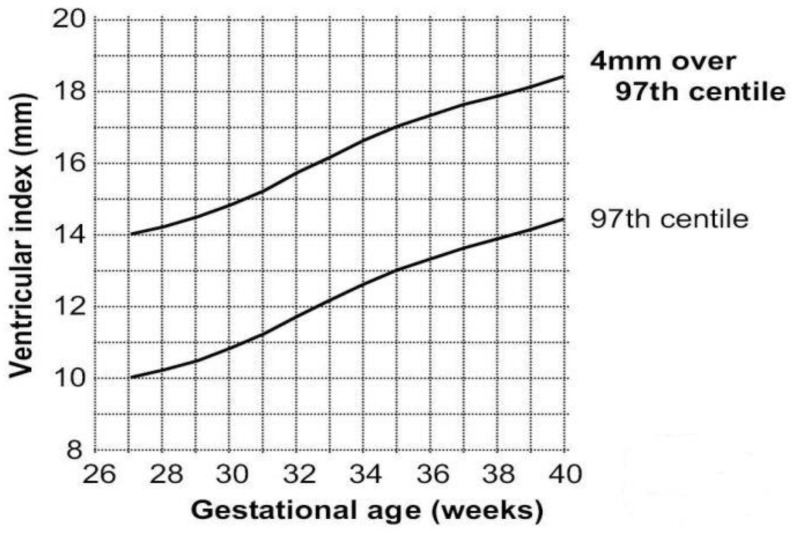
The 97th and 97th + 4 mm lines of VI at different gestational ages (source: Levene et al. [9] Arch Dis Child. 1981, 56: 900–904; VI: ventricular index).

**Figure 4 jcm-11-07468-f004:**
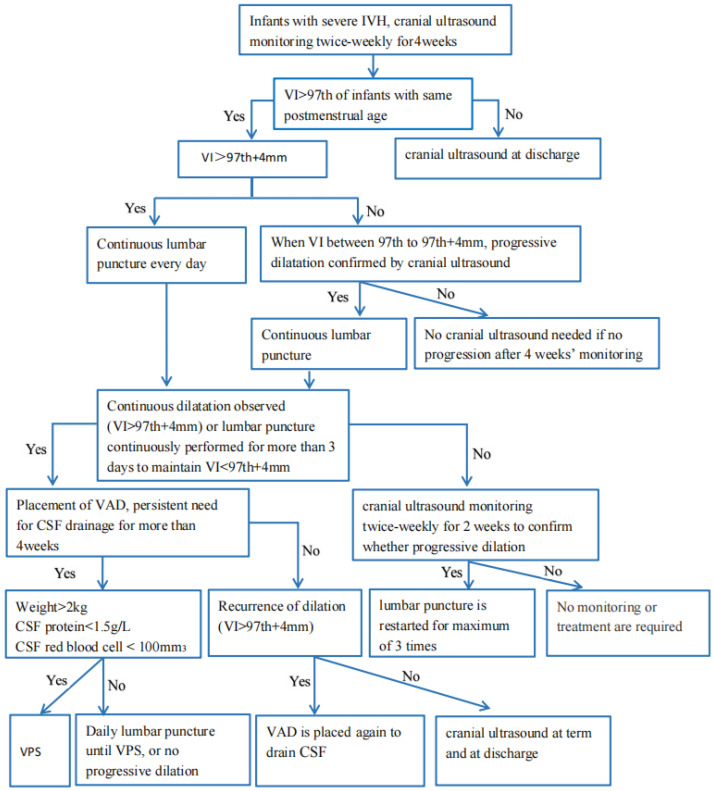
Flow diagram of monitoring and treatment of PHVD (IVH: intraventricular hemorrhage; VI: ventricular index; VAD: ventricular access device; CSF: cerebrospinal fluid; VPS: ventriculoperitoneal shunt).

## Data Availability

All data is included in the manuscript.

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
