# Peer review of "Ultrasonic Diagnosis and Management of Posthemorrhagic Ventricular Dilatation in Premature Infants: A Narrative Review"

_jcm, 2022, doi:10.3390/jcm11247468_

Round 1
Reviewer 1 Report
Liu and Nie have presented an excellent summary of the management of posthemorrhagic hydrocephalus, an area that requires improved standardization and examination. The presentation is limited by the lack of discussion of the controversial literature that often characterizes this topic. While limited space may have contributed to this approach, still a recognition of this issue has merit. Finally, the lack of correlation between changes in head circumference and ventricular dilatation, as note by Volpe, is an important observation that may influence management, and should be discussed.
Reviewer 2 Report
This is an interesting review manuscript covering an important and relevant topic about the survival of preterm infants with intraventricular hemorrhage. While the authors provide a lot of useful information, the manuscript needs to be organized as a review manuscript. I have some suggestions on how to improve the manuscript:
1. Title:
It is unclear whether this manuscript is a review manuscript and this should be mentioned in the title.
2. Abstract:
Line 10: I would suggest including the full term Volpe Grades III-IV.
3. Introduction:
The format or structure of a narrative review should include: Introduction, Methods, Results, Discussion.
Line 19: Rephrase to: "Because of the fragile structure of the germinal matrix...."
Line 20: Does the term "preterm infants" include all infants <37 weeks gestation. How about very preterm/extremely preterm infants? This should be clarified.
Line 28: I would suggest including the term Volpe Grades III-IV.
Line 29: Which is the main reason of neurodevelopmental....?
Search strategy? Databases and keywords used? Inclusion/exclusion criteria?
Lines 31-32: Rephrase to "Treatment of PHVD in preterm infants...."
Line 36: Used to monitor what? PVHD? This sentence is unclear.
Figure 1 and Figure 2: What is the source of these figures? If they were taken from another article, it should be cited.
Lines 69-71: "It is reported that"....this sentence is unclear. Do the authors mean after clinical symptoms are present? This should be rephrased.
Lines 77-80: What is the source/citation of this information? I would suggest that the authors provide a source/citation for these statements.
Line 96: "before neurosurgical intervention thus..." this sentence is unclear and needs to be expanded.
Lines 98-103. These statements need a source/citation. Where did the authors get this information from?
Line 107: Do the author's mean statistically significant difference? It is unclear from this statement.
Line 122: The citation format should be consistent throughout the entire manuscript. Therefore, the citation (superscript 9) should be in brackets [9] as are the other citations in the manuscript.
Line 127: should be a source/citation after word "Ommaya capsule." Also, the authors should explain what is an Ommaya capsule to readers who are not familiar with pediatric neurosurgery.
Line 146: Rephrase to "has elasticity and absorption capacity".
Line 173-177: I would suggest to include the definition of a VPS and explain what it is and how it functions, and include a source/citation of the information.
Lines 173-177: What is the source of this information> The authors need to provide citations.
Line 178: "Some PHVD can spontaneously stop progressing....." this statement is unclear. Do the authors mean some types of PHVD or PHVD in general? This needs to be clarified.
Lines 180-181: "A VPS transfers the excess..." I would move this sentence higher up to lines 174-175.
Line 190: "Infection and device damage are the most important complications..." This sentence needs a source/citation.
Lines 191-193: "If an infant with a drainage device develops a persistent fever...." this sentence is not entirely clear and needs to be rephrased.
4. Conclusions:
What is the summary of the evidence? Limitations? Implications for future research?
Line 200: "A VPS should be placed if the weight >2kg...." This statement is unclear. Do the authors mean weight of the infant?
Line 202: "Based on commonly used management schemes....."
Lines 204-205: And way to intervene?
Round 2
Reviewer 2 Report
Thank you for revising your manuscript according to my comments and suggestions. I appreciate your effort in improving the quality of this interesting manuscript.
I only have a few minor suggestions/edits:
Title: The word "Narrative Review" should be included in the title of the manuscript.
Introduction: I would suggest not to begin the Introduction with the word "Because". This sentence should rephrased.
Line 50: I would suggest to mention that this is a narrative literature review.
Lines 50-52: I would suggest to rephrase to "The purpose of this report is to review the literature for diagnosis and therapeutic strategies of PVHD".
Figure 1 and Figure 2: I would suggest that the authors mention that the figures were from their own resident patients. This information needs to be written under the figures.
